# EEG-based detection of the locus of auditory attention with convolutional neural networks

**Servaas Vandecappelle[1,2]\*, Lucas Deckers[1,2], Neetha Das[1,2], Amir Hossein Ansari[2], Alexander Bertrand[2], Tom Francart[1]\***

[1]Department of Neurosciences, Experimental Oto-rhino-laryngology, Leuven, Belgium; [2]Department of Electrical Engineering (ESAT), Stadius Center for Dynamical Systems, Signal Processing and Data Analytics, Leuven, Belgium

**Abstract** In a multi-speaker scenario, the human auditory system is able to attend to one particular speaker of interest and ignore the others. It has been demonstrated that it is possible to use electroencephalography (EEG) signals to infer to which speaker someone is attending by relating the neural activity to the speech signals. However, classifying auditory attention within a short time interval remains the main challenge. We present a convolutional neural network-based approach to extract the locus of auditory attention (left/right) without knowledge of the speech envelopes. Our results show that it is possible to decode the locus of attention within 1–2 s, with a median accuracy of around 81%. These results are promising for neuro-steered noise suppression in hearing aids, in particular in scenarios where per-speaker envelopes are unavailable.

## Introduction

In a multi-speaker scenario, the human auditory system is able to focus on just one speaker, ignoring all other speakers and noise. This situation is called the 'cocktail party problem' (*Cherry, 1953*). However, elderly people and people suffering from hearing loss have particular difficulty attending to one person in such an environment. In current hearing aids, this problem is mitigated by automatic noise suppression systems. When multiple speakers are present, however, these systems have to rely on heuristics such as the speaker volume or the listener's look direction to determine the relevant speaker, which often fail in practice.

The emerging field of auditory attention decoding (AAD) tackles the challenge of directly decoding auditory attention from neural activity, which may replace such unreliable and indirect heuristics. This research finds applications in the development of neuro-steered hearing prostheses that analyze brain signals to automatically decode the direction or speaker to whom the user is attending, to subsequently amplify that specific speech stream while suppressing other speech streams and surrounding noise. The desired result is increased speech intelligibility for the listener.

In a competing two-speaker scenario, it has been shown that the neural activity (as recorded using electroencephalography [EEG] or magnetoencephalography [MEG]) consistently tracks the dynamic variation of an incoming speech envelope during auditory processing, and that the attended speech envelope is typically more pronounced than the unattended speech envelope (*Ding and Simon, 2012*; *O'Sullivan et al., 2015*). This neural tracking of the stimulus can then be used to determine auditory attention. A common approach is stimulus reconstruction, where the poststimulus brain activity is used to decode and reconstruct the attended stimulus envelope (*O'Sullivan et al., 2015*; *Pasley et al., 2012*). The reconstructed envelope is then correlated with the original stimulus envelopes, and the one yielding the highest correlation is then considered to belong to the attended speaker. Other methods for attention decoding include the forward

**\*For correspondence:**
servaas.vandecappelle@gmail.com (SV);
tom.francart@kuleuven.be (TF)

**Competing interests:** The authors declare that no competing interests exist.

modeling approach: predicting EEG from the auditory stimulus (*Akram et al., 2016*; *Alickovic et al., 2016*), canonical correlation analysis (CCA)-based methods (*de Cheveigné et al., 2018*), and Bayesian state-space modeling (*Miran et al., 2018*).

All studies mentioned above are based on linear decoders. However, since the human auditory system is inherently nonlinear (*Faure and Korn, 2001*), nonlinear models (such as neural networks) could be beneficial for reliable and quick AAD. In *Taillez et al., 2017*, a feedforward neural network for EEG-based speech stimulus reconstruction was presented, showing that artificial neural networks are a feasible alternative to linear decoding methods.

Recently, convolutional neural networks (CNNs) have become the preferred approach for many recognition and detection tasks, in particular in the field of image classification (*LeCun et al., 2015*). Recent research on CNNs has also shown promising results for EEG classification: in seizure detection (*Acharya et al., 2018a*; *Ansari et al., 2018a*), depression detection (*Liu et al., 2017*), and sleep stage classification (*Acharya et al., 2018b*; *Ansari et al., 2018b*). In terms of EEG-based AAD, *Ciccarelli et al., 2019* recently showed that a (subject-dependent) CNN using a classification approach can outperform linear methods for decision windows of 10 s.

Current state-of-the-art models are thus capable of classifying auditory attention in a two-speaker scenario with high accuracy (75–85%) over a data window with a length of 10 s, but their performance drops drastically when shorter windows are used (e.g., *de Cheveigné et al., 2018*; *Ciccarelli et al., 2019*). However, to achieve sufficiently fast AAD-based steering of a hearing aid, short decision windows (down to a few seconds) are required. This inherent trade-off between accuracy and decision window length was investigated by *Geirnaert et al., 2020*, who proposed a method to combine both properties into a single metric, by searching for the optimal trade-off point to minimize the expected switch duration in an AAD-based volume control system with robustness constraints. The robustness against AAD errors can be improved by using smaller relative volume changes for every new AAD decision, while the decision window length determines how often an AAD decision (volume step) is made. It was found that such systems favor short window lengths (<< 10 s) with mediocre accuracy over long windows (10–30 s) with high accuracy.

Apart from decoding which speech envelope corresponds to the attended speaker, it may also be possible to decode the spatial locus of attention. That is, not decoding which *speaker* is attended to, but rather which location in space. The benefit of this approach for neuro-steered auditory prostheses is that no access to the clean speech stimuli is needed. This has been investigated based on differences in the EEG entropy features (*Lu et al., 2018*), but the performance was insufficient for practical use (below 70% for 60 s windows). However, recent research (*Wolbers et al., 2011*; *Bednar and Lalor, 2018*; *Patel et al., 2018*; *O'Sullivan et al., 2019*; *Bednar and Lalor, 2020*) has shown that the direction of auditory attention is neurally encoded, indicating that it could be possible to decode the attended sound position or trajectory from EEG. A few studies employing MEG have suggested that in particular the alpha power band could be tracked to determine the locus of auditory attention (*Frey et al., 2014*; *Wöstmann et al., 2016*). Another study, employing scalp EEG, found the beta power band related with selective attention (*Gao et al., 2017*).

The aim of this paper is to further explore the possibilities of CNNs for EEG-based AAD. As opposed to *Taillez et al., 2017* and *Ciccarelli et al., 2019*, who aim to decode the attended speaker (for a given set of speech envelopes), we aim to decode the locus of auditory attention (left/right). When the locus of attention is known, a hearing aid can steer a beamformer in that direction to enhance the attended speaker.

## Materials and methods

### Experiment setup

The dataset used for this work was gathered previously (*Das et al., 2016*). EEG data was collected from 16 normal-hearing subjects while they listened to two competing speakers and were instructed to attend to one particular speaker. Every subject signed an informed consent form approved by the KU Leuven ethical committee.

The EEG data was recorded using a 64-channel BioSemi ActiveTwo system, at a sampling rate of 8196 Hz, in an electromagnetically shielded and soundproof room. The auditory stimuli were low-

pass filtered with a cutoff frequency of 4 kHz and presented at 60 dBA through Etymotic ER3 insert earphones. APEX 3 was used as stimulation software (*Francart et al., 2008*).

The auditory stimuli were comprised of four Dutch stories, narrated by three male Flemish speakers (*DeBuren, 2007*). Each story was 12 min long and split into two parts of 6 min each. Silent segments longer than 500 ms were shortened to 500 ms. The stimuli were set to equal root-mean-square intensities and were perceived as equally loud.

The experiment was split into eight trials, each 6 min long. In every trial, subjects were presented with two parts of two different stories. One part was presented in the left ear, while the other was presented in the right ear. Subjects were instructed to attend to one of the two via a monitor positioned in front of them. The symbol '<' was shown on the left side of the screen when subjects had to attend to the story in the left ear, and the symbol '>' was shown on the right side of the screen when subjects had to attend to the story in the right ear. They did not receive instructions on where to focus their gaze.

In subsequent trials, subjects attended either to the second part of the same story (so they could follow the story line) or to the first part of the next story. After each trial, subjects completed a multiple-choice quiz about the attended story. In total, there was 8 × 6 min = 48 min of data per subject. For an example of how stimuli were presented, see *Table 1*. (The original experiment [*Das et al., 2016*] contained 12 additional trials of 2 min each, collected at the end of every measurement session. These trials were repetitions of earlier stimuli and were not used in this work.)

The attended ear alternated over consecutive trials to get an equal amount of data per ear (and per subject), which is important to avoid the lateralization bias described by *Das et al., 2016*. Stimuli were presented in the same order to each subject, and either dichotically or after head-related transfer function (HRTF) filtering (simulating sound coming from ±90 deg). As with the attended ear, the HRTF/dichotic condition was randomized and balanced within and over subjects. In this work, we do not distinguish between dichotic and HRTF to ensure there is as much data as possible for training the neural network.

## Data preprocessing

The EEG data was filtered with an equiripple FIR bandpass filter and its group delay was compensated for. For use with linear models, the EEG was filtered between 1 and 9 Hz, which has been found to be an optimal frequency range for linear attention decoding (*Pasley et al., 2012*; *Ding and Simon, 2012*). For the CNN models, a broader bandwidth between 1 and 32 Hz was used, as *Taillez et al., 2017* show that this is more optimal. In both cases, the maximal bandpass attenuation was 0.5 dB while the stopband attenuation was 20 dB (at 0–1 Hz) and 15 dB (at 32–64 Hz). After the bandpass filtering, the EEG data was downsampled to 20 Hz (linear model) and 128 Hz (CNN). Artifacts were removed with the generic MWF-based removal algorithm described in *Somers et al., 2018*.

**Table 1.** First eight trials for a random subject.

Trials are numbered according to the order in which they were presented to the subject. Which ear was attended to first was determined randomly. After that, the attended ear was alternated. Presentation (dichotic/HRTF) was balanced over subjects with respect to the attended ear. Adapted from *Das et al., 2016*. HRTF = head-related transfer function.

| Trial | Left stimulus | Right stimulus | Attended ear | Presentation |
|-------|---------------|----------------|--------------|--------------|
| 1 | Story1, part1 | Story2, part1 | Left | Dichotic |
| 2 | Story2, part2 | Story1, part2 | Right | HRTF |
| 3 | Story3, part1 | Story4, part1 | Left | Dichotic |
| 4 | Story4, part2 | Story3, part2 | Right | HRTF |
| 5 | Story2, part1 | Story1, part1 | Left | Dichotic |
| 6 | Story1, part2 | Story2, part2 | Right | HRTF |
| 7 | Story4, part1 | Story3, part1 | Left | Dichotic |
| 8 | Story3, part2 | Story4, part2 | Right | HRTF |

Data of each subject was divided into a training, validation, and test set. Per set, data segments were generated with a sliding window equal in size to the chosen window length and with an overlap of 50%. Data was normalized on a subject-by-subject basis, based on statistics of the training set only, and in such a way that proportions between EEG channels were maintained. Concretely, for each subject we calculated the power per channel, based on the 10% trimmed mean of the squared samples. All channels were then divided by the square root of the median of those 64 values (one for each EEG channel). Data of each subject was thus normalized based on a single (subject-specific) value.

## Convolutional neural networks

A convolutional neural network (CNN) consists of a series of convolutional layers and nonlinear activation functions, typically followed by pooling layers. In convolutional layers, one or more convolutional filters slide over the data to extract local data features. Pooling layers then aggregate the output by computing, for example, the mean. Similar to other types of neural networks, a CNN is optimized by minimizing a loss function, and the optimal parameters are estimated with an optimization algorithm such as stochastic gradient descent.

Our proposed CNN for decoding the locus of auditory attention is shown in *Figure 1*. The input is a $64 \times T$ matrix, where 64 is the number of EEG channels in our dataset and $T$ is the number of samples in the decision window. (We tested multiple decision window lengths, as discussed later.) The first step in the model is a convolutional layer, indicated in blue. Five independent $64 \times 17$ spatio-temporal filters are shifted over the input matrix, which, since the first dimension is equal to the number of channels, each result in a time series of dimensions $1 \times T$. Note that '17' is 130 ms at 128 Hz, and 130 ms was found to be an optimal filter width – that is, longer or shorter decision window lengths gave a higher loss on a validation set. A rectifying linear unit (ReLu) activation function is used after the convolution step.

In the average pooling step, data is averaged over the time dimension, thus reducing each time series to a single number. After the pooling step, there are two fully connected (FC) layers. The first layer contains five neurons (one for each time series) and is followed by a sigmoid activation function, and the second layer contains two (output) neurons. These two neurons are connected to a cross-entropy loss function. Note that with only two directions (left/right), a single output neuron (coupled with a binary cross-entropy loss) would have sufficed as well. With this setup, it is easier to extend to more locations, however. The full CNN consists of approximately 5500 parameters.

The implementation was done in MATLAB 2016b and MatConvNet (version 1.0-beta25), a CNN toolbox for MATLAB (*Vedaldi and Lenc, 2015*). The source code is available at https://github.com/

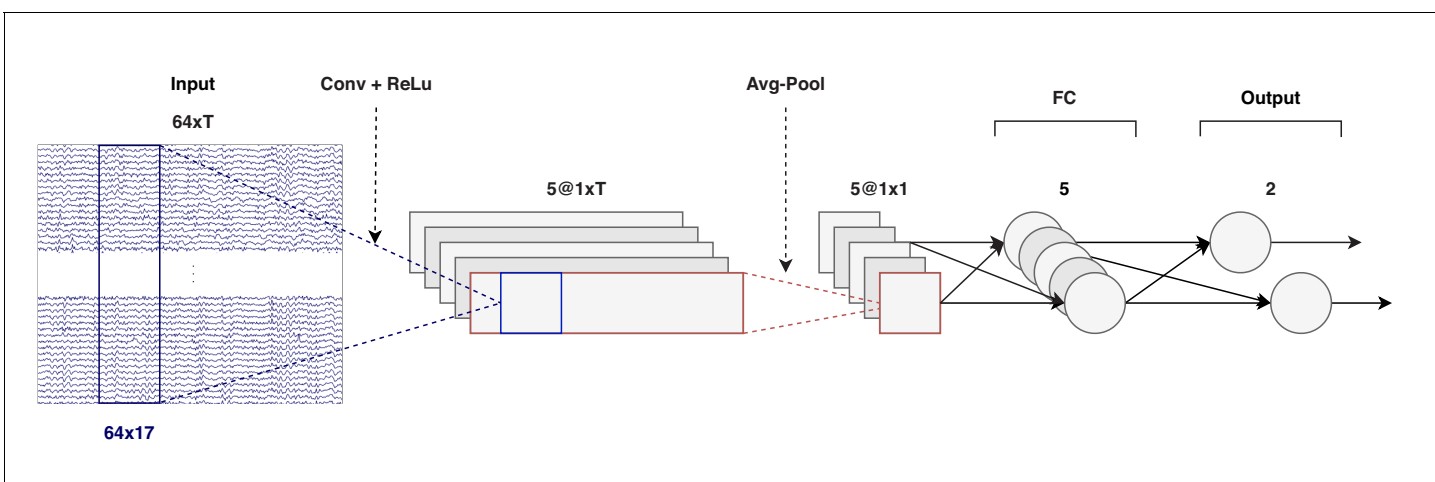

**Figure 1.** CNN architecture (windows of $T$ samples). Input: $T$ time samples of a 64-channel EEG signal, at a sampling rate of 128 Hz. Output: two scalars that determine the attended direction (left/right). The convolution, shown in blue, considers 130 ms of data over all channels. EEG = electroencephalography, CNN = convolutional neural network, ReLu = rectifying linear unit, FC = fully connected.

exporl/locus-of-auditory-attention-cnn (copy archived at swh:1:rev:3e5e21a7e6072182e076-f9863ebc82b85e7a01b1; *Vandecappelle, 2021*).

## CNN training and evaluation

The model was trained on data of all subjects, including the subject it was tested on (but without using the same data for both training and testing). This means we are training a subject-specific decoder, where the data of the other subjects can be viewed as a regularization or data augmentation technique to avoid overfitting on the (limited) amount of training data of the subject under test.

To prevent the model from overfitting to one particular story, we cross-validated over the four stories (resulting in four folds). That is, we held out one story and trained on the remaining three stories (illustrated in *Table 2*). Such overfitting is not an issue for simple linear models, but may be an issue for the CNN we propose here. Indeed, even showing only the EEG responses to a part of a story could result in the model learning certain story-specific characteristics. That could then lead to overly optimistic results when the model is presented with the EEG responses to another (albeit different) part of the same story. Similarly, since each speaker has their own 'story-telling' characteristics (e.g., speaking rate or intonation), and a different voice timbre, EEG responses to different speakers may differ. Therefore, it is possible that the model gains an advantage by having 'seen' the EEG response to a specific speaker, so we retained only the folds wherein the same speaker was never simultaneously part of both the training and the test set. In the end, only two folds remained (see *Table 2*). We refer to the combined cross-validation approach as *leave-one-story+speaker-out*.

In an additional experiment, we investigated the subject dependency of the model, where, in addition to cross-validating over story and speaker, we also cross-validated over subjects. That is, we no longer trained and tested on $N$ subjects, but instead trained on $N - 1$ subjects and tested on the held-out subject. Such a paradigm has the advantage that new subjects do not have to undergo potentially expensive and time-consuming retraining, making it more suitable for real-life applications. Whether it is actually a better choice than subject-specific retraining depends on the difference in performance between the two paradigms. If the difference is sufficiently large, subject-dependent retraining might be a price one is willing to pay.

We trained the network by minimizing the cross-entropy between the network outputs and the corresponding labels (the attended ear). We used mini-batch stochastic gradient descent with an initial learning rate of 0.09 and a momentum of 0.9. We applied a step decay learning schedule that decreased the learning rate after epoch 10 and 35 to 0.045 and 0.0225, respectively, to assure convergence. The batch size was set to 20, partly because of memory constraints, and partly because we did not see much improvement with larger batch sizes. Weights and biases were initialized by drawing randomly from a normal distribution with a mean of 0 and a standard deviation of 0.5.

**Table 2.** Cross-validating over stories and speakers.

With the current dataset, there are only two folds that do not mix stories and speakers across training and test sets. Top: Story 1 as test data; story 2, 3, and 4 as training data and validation data (85/15% division, per story). Bottom: similarly, but now with a different story and speaker as test data. In both cases, the story and speaker are completely unseen by the model. The model is trained on the same training set for all subjects and tested on a unique, subject-specific, test set.

| Story | Speaker | Subject 1 | Subject 2 | . . . | Subject 16 |
|-------|---------|-----------|-----------|-------|------------|
| 1 | 1 | test | test | . . . | test |
| 2 | 2 | train/val | | | |
| 3 | 3 | train/val | | | |
| 4 | 3 | train/val | | | |
| Story | Speaker | Subject 1 | Subject 2 | . . . | Subject 16 |
| 1 | 1 | train/val | | | |
| 2 | 2 | test | test | . . . | test |
| 3 | 3 | train/val | | | |
| 4 | 3 | train/val | | | |

Training ran for 100 epochs, as early experiments showed that the optimal decoder was usually found between epoch 70 and 95. Regularization consisted of weight decay with a value of $5 \times 10^{-4}$, and, after training, of selecting the decoder in the iteration where the validation loss was minimal. Note that the addition of data of the other subjects can also be viewed as a regularization technique that further reduces the risk of overfitting.

All hyperparameters given above were determined by running a grid search over a set of reasonable values. Performance during this grid search was measured on the validation set.

Note that in this work the decoding accuracy is defined as the percentage of correctly classified decision windows on the test set, averaged over the two folds mentioned earlier (one for each story narrated by a different speaker).

### Linear baseline model (stimulus reconstruction)

A linear stimulus reconstruction model (*Biesmans et al., 2017*) was used as baseline. In this model, a spatio-temporal filter was trained and applied on the EEG data and its time-shifted versions up to 250 ms delay, based on least-squares regression, in order to reconstruct the envelope of the attended stimulus. The reconstructed envelope was then correlated (Pearson correlation coefficient) with each of the two speaker envelopes over a data window with a predefined length, denoted as the decision window (different lengths were tested). The classification was made by selecting the position corresponding to the speaker that yielded the highest correlation in this decision window. The envelopes were calculated with the 'powerlaw subbands' method proposed by *Biesmans et al., 2017*; that is, a gammatone filter bank was used to split the speech into subbands, and per subband the envelope was calculated with a power law compression with exponent 0.6. The different subbands were then added again (each with a coefficient of 1) to form the broadband envelope. Envelopes were filtered and downsampled in the same vein as the EEG recordings.

For a fairer comparison with the CNN, the linear model was also trained in a *leave-one-story +speaker-out* way. In contrast to the CNN, however, the linear model was not trained on any other data than that of the subject under testing, since including data of other subjects harms the performance of the linear model.

Note that the results of the linear model here merely serve as a representative baseline, and that a comparison between the two models should be treated with care – in part because the CNN is nonlinear, but also because the linear model is only able to relate the EEG to the envelopes of the recorded audio, while the CNN is free to extract any feature it finds optimal (though only from the EEG, as no audio is given to the CNN). Additionally, the preprossessing is slightly different for both models. However, that preprocessing was chosen such that each model would perform optimally – using the same preprocessing would, in fact, negatively impact one of the two models.

### Minimal expected switch duration

For some of the statistical tests below, we use the minimal expected switch duration (MESD) proposed by *Geirnaert et al., 2020* as a relevant metric to assess AAD performance. The goal of the MESD metric is to have a single value as measure of performance, resolving the trade-off between accuracy and the decision window length. The MESD was defined as the expected time required for an AAD-based gain control system to reach a stable volume switch between both speakers, following an attention switch of the user. The MESD is calculated by optimizing a Markov chain as a model for the volume control system, which uses the AAD decision time and decoding accuracy as parameters. As a by-product, it provides the optimal volume increment per AAD decision.

One caveat is that the MESD metric assumes that all decisions are taken independently of each other, but this may not be true when the window length is very small, for example, smaller than 1 s. In that case, the model behind the MESD metric may slightly underestimate the time needed for a stable switch to occur. However, it can still serve as a useful tool for comparing models.

## Results

### Decoding performance

Seven different decision window lengths were tested: 10, 5, 2, 1, 0.5, 0.25, and 0.13 s. This defines the amount of data that is used to make a single left/right decision. In the AAD literature, decision

windows range from approximately 60 to 5 s. In this work, the focus lies on shorter decision windows. This is done for practical reasons: in neuro-steered hearing aid applications, the detection time should ideally be short enough to quickly detect attention switches of the user.

To capture the general performance of the CNN, the reported accuracy for each subject is the mean accuracy of 10 different training runs of the model, each with a different (random) initialization. All MESD values in this work are based on these mean accuracies.

The linear model was not evaluated at a decision window length of 0.13 s since its kernel has a width of 0.25 s, which places a lower bound on the possible decision window length.

*Figure 2* shows the decoding accuracy at 1 and 10 s for the CNN and the linear model. For both decision windows, the CNN had a higher median decoding accuracy, but a larger intersubject variability. Two subjects had a decoding accuracy lower than 50% at a window length of 10 s, and were therefore not considered in the subsequent analysis, nor are they shown in the figures in this section.

For 1 s decision windows, a Wilcoxon signed-rank test yielded significant differences in detection accuracy between the linear decoder model and the CNN ($W = 3$, $p < 0.001$), with an increase in median accuracy from 58.1 to 80.8%. Similarly, for 10 s decision windows, a Wilcoxon signed-rank test showed a significant difference between the two models ($W = 16$, $p = 0.0203$), with the CNN achieving a median accuracy of 85.1% compared to 75.7% for the linear model.

The minimal expected switch duration (MESD) (*Geirnaert et al., 2020*) outputs a single number for each subject, given a set of window lengths and corresponding decoding accuracies. This allows for a direct comparison between the linear and the CNN model, independent of window length. As shown in *Figure 3*, the linear model achieves a median MESD of 22.6 s, while the CNN achieves a median MESD of only 0.819 s. A Wilcoxon signed-rank test shows this difference to be significant ($W = 105$, $p < 0.001$). The extremely low MESD for the CNN is the result of the median accuracy still being 68.7% at only 0.13 s, and the fact that the MESD typically chooses the optimal operation point at short decision window lengths (*Geirnaert et al., 2020*).

It is not entirely clear why the CNN fails for 2 of the 16 subjects. Our analysis shows that the results depend heavily on the story that is being tested on: for the two subjects with below 50% accuracy, the CNN performed poorly on story 1 and 2, but performed well on stories 3 and 4 (80% and higher). Our results are based on stories 1 and 2, however, since stories 3 and 4 are narrated by the same speaker and we wanted to avoid having the same speaker in both the training and test set. It is possible that the subjects did not comply with the task in these conditions.

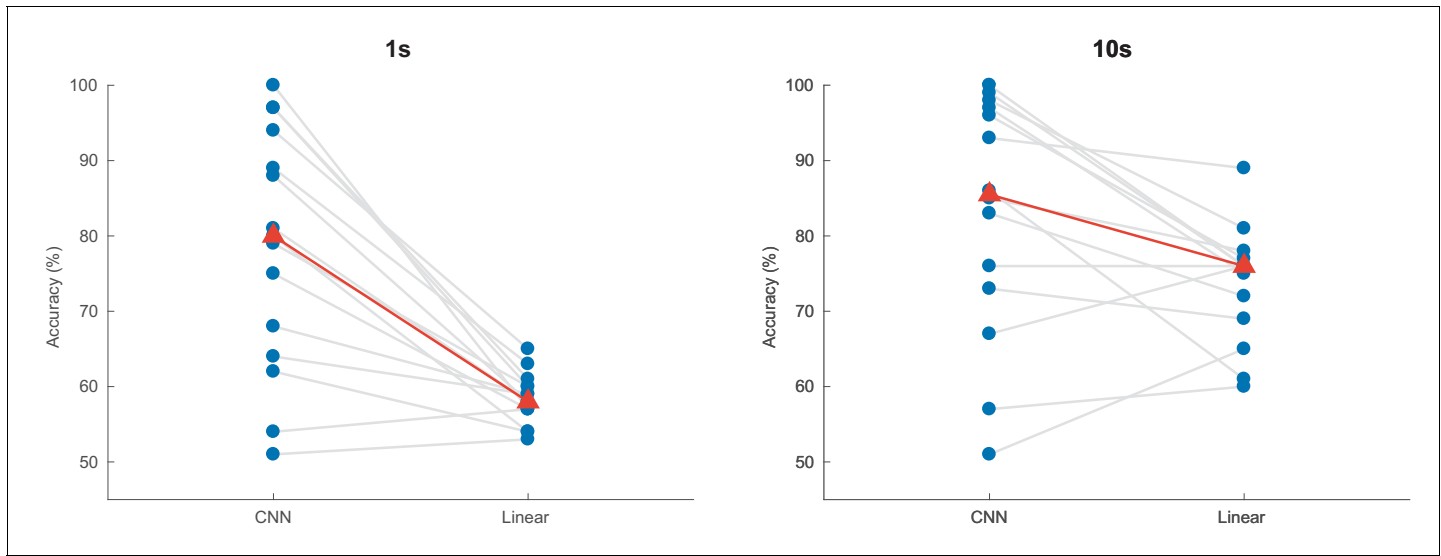

**Figure 2.** Auditory attention detection performance of the CNN for two different window lengths. Linear decoding model shown as baseline. Blue dots: per-subject results, averaged over two test stories. Gray lines: same subjects. Red triangles: median accuracies. CNN = convolutional neural network.

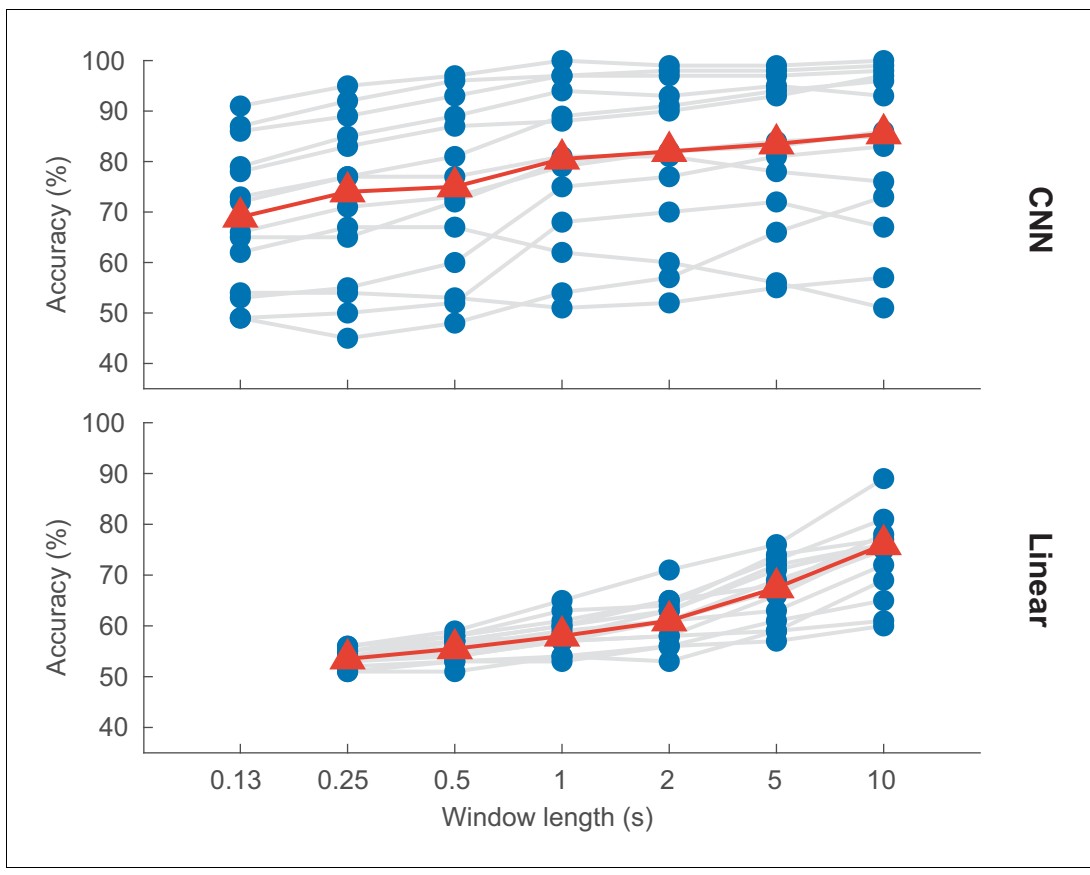

**Figure 3.** Minimal expected switch durations (MESDs) for the CNN and the linear baseline. Dots: per-subject results, averaged over two test stories. Gray lines: same subjects. Vertical black bars: median MESD. As before, two poorly performing subjects were excluded from the analysis. CNN = convolutional neural network.

## Effect of decision window length

Shorter decision windows contain less information and should therefore result in poorer performance compared to longer decision windows. *Figure 4* visualizes the relation between window length and detection accuracy.

A linear mixed-effects model fit for decoding accuracy, with decision window length as fixed effect and subject as random effect, shows a significant effect of window length for both the CNN model (*df* = 96, *p* < 0.001) and the linear model (*df* = 94, *p* < 0.001). The analysis was based on the decision window lengths shown in *Figure 4*; that is, seven window lengths for the CNN and six for the linear model.

**Figure 4.** Auditory attention detection performance as a function of the decision window length. Blue dots: per-subject results, averaged over two test stories. Gray lines: same subjects. Red triangles: median accuracies. CNN = convolutional neural network.

## Interpretation of results

Interpreting the mechanisms behind a neural network remains a challenge. In an attempt to understand which frequency bands of the EEG the network uses, we retested (without retraining) the model in two ways: (1) by filtering out a certain frequency range (*Figure 5*, left); (2) by filtering out everything *except* a particular frequency range (*Figure 5*, right). The frequency ranges are defined as follows: δ = 1–4 Hz; θ = 4–8 Hz; α = 8–14 Hz; β = 14–32 Hz.

*Figure 5* shows that the CNN uses mainly information from the beta band, in line with *Gao et al., 2017*. Note that the poor results for the other frequency bands (*Figure 5*, right) does not necessarily mean that the network does not use the other bands, but rather, if it does, it is in combination with other bands.

We additionally investigated the weights of the filters of the convolutional layer, as they give an indication of what channel the model finds important. We calculated the power of the filter weights per channel, and to capture the general trend, we calculated a grand-average over all models (i.e., all window lengths, stories, and runs). Moreover, we normalized the results with the per-channel power of the EEG in the training set, to account for that fact that what comes out of the convolutional layer is a function of both the filter weights and the magnitude of the input.

The results are shown in *Figure 6*. We see primarily activations in the frontal and temporal regions, and to a lesser extent also in the occipital lobe. Activations appear to be slightly stronger on the right side, as well. This result is in line with *Ciccarelli et al., 2019*, who also saw stronger activations in the frontal channels (mostly for the 'Wet 18 CH' and 'Dry 18 CH' systems). Additionally, *Gao et al., 2017* also found the frontal channels to significantly differ from the other channels within the beta band (Figure 3 and Table 1 in *Gao et al., 2017*). The prior (eye) artifact removal step in the EEG preprocessing and the importance of the beta band in the decision-making (*Figure 5*) suggests that the focus on the frontal channel is not necessarily attributed to eye artifacts. It is noted that the filters of the network act as backward decoders, and therefore care should be taken when interpreting topoplots related to the decoder coefficients. As opposed to a forward (encoding) model, the coefficients of a backward (decoding) model are not necessarily predictive for the strength of the neural response in these channels. For example, the network may perform an implicit noise reduction transformation, thereby involving channels with low SNR as well.

## Effect of validation procedure

In all previous results, we used a *leave-one-story+speaker-out* scheme to prevent the CNN from gaining an advantage by already having seen EEG responses elicited by the same speaker or

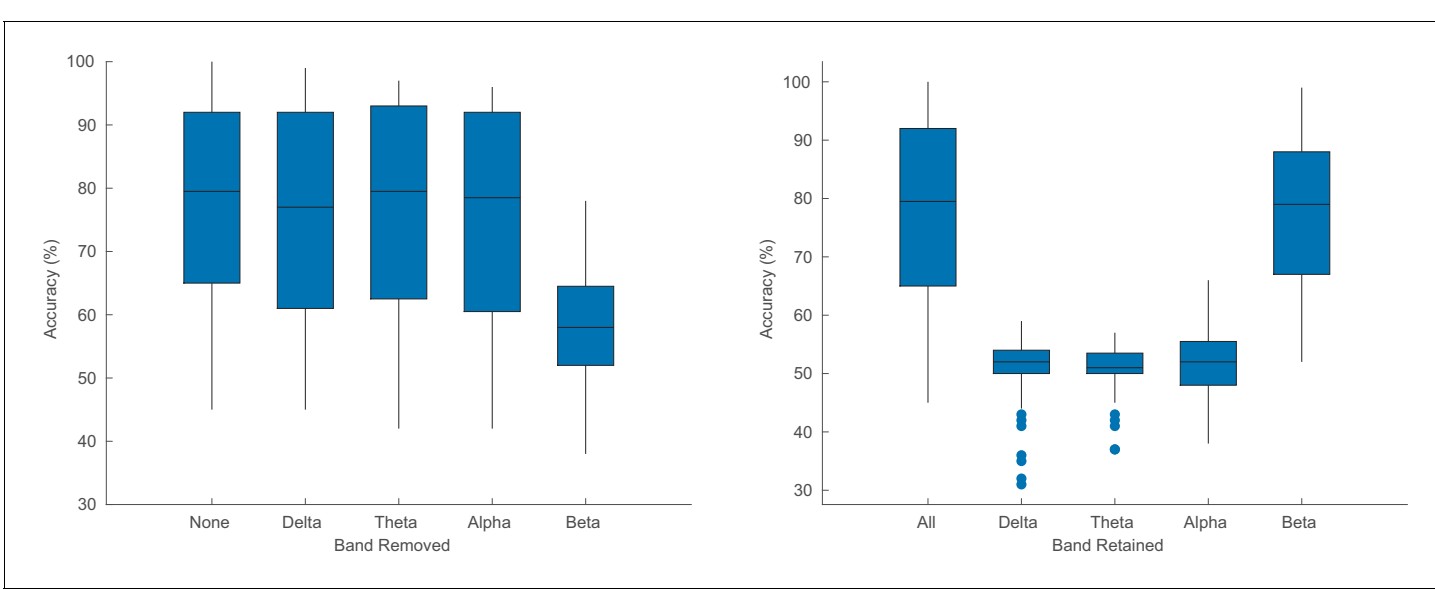

**Figure 5.** Auditory attention detection performance of the CNN when one particular frequency band is removed (left) and when only one band is used (right). The original results are also shown for reference. Each box plot contains results for all window lengths and for the two test stories.

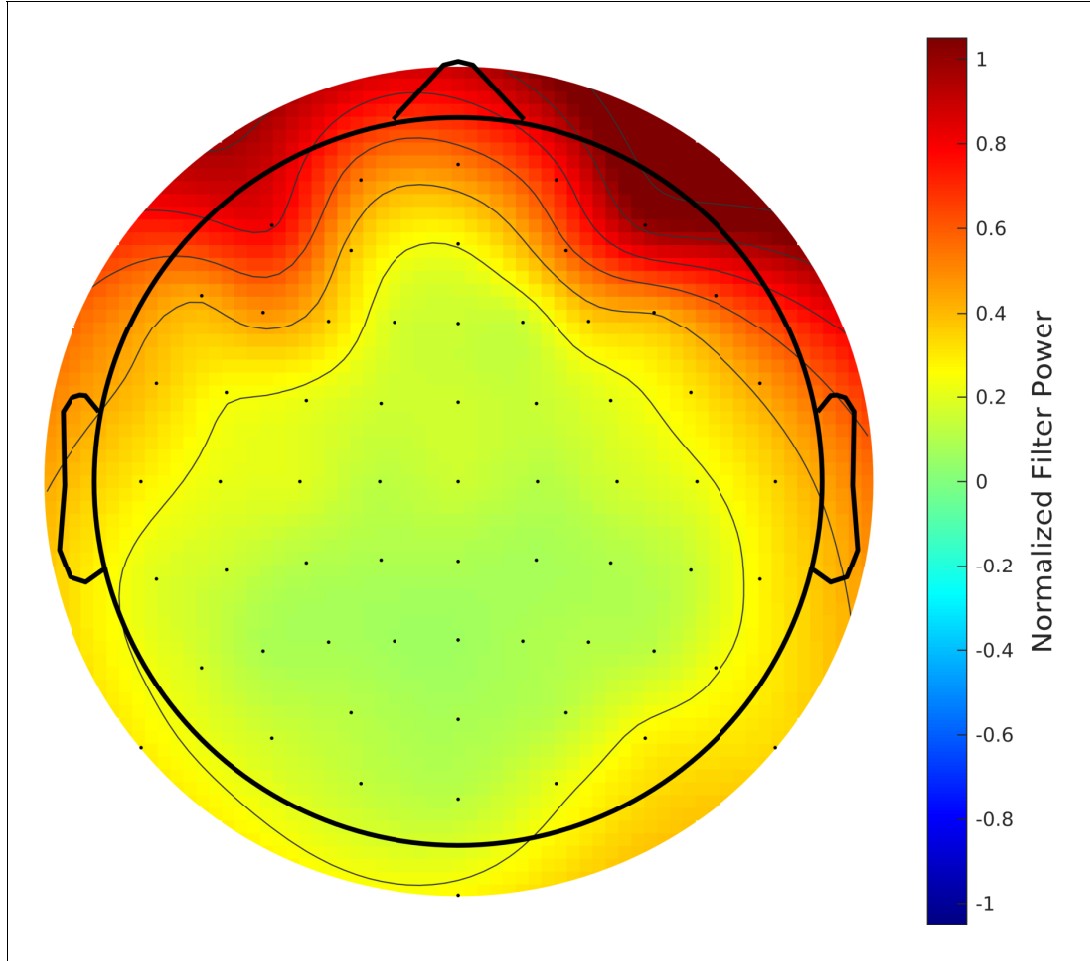

**Figure 6.** Grand-average topographic map of the normalized power of convolutional filters.

different parts of the same story. However, it is noted that in the majority of the AAD literature, training and test sets often do contain samples from the same speaker or story (albeit from different parts of the story).

To investigate the impact of cross-validating over speaker and story, we trained the CNN again, but this time using data of each trial (later referred to as 'Every trial'). Here, the training set consisted of the first 75% of each trial, the validation set of the next 15% and the test set of the last 15%. We performed this experiment twice – once using data preprocessed in the manner explained in the ''Data processing'' section, and once with the artefact removal filtering (MWF) stage excluded.

*Figure 7* shows the results of all three experiments for decision windows of 1 s. Other window lengths show similar results.

For decision windows of 1 s, using data from all trials, in addition to applying a per-trial MWF filter, results in a median decoding accuracy of 92.8% (*Figure 7*, right), compared to only 80.8% when leaving out both story and speaker (*Figure 7*, left). A Wilcoxon signed-rank test shows this difference to be significant ($W = 91$, $p = 0.0134$). There is, however, no statistically significant difference in decoding accuracy between leaving out both story and speaker and when using data of all trials, but without applying any spatial filtering for artifact removal ($W = 48$, $p = 0.8077$).

It appears that having the same speaker and story in both the training and test set is less problematic than we had anticipated, and employing a classical scheme wherein both sets draw from the same trials (though use different parts) is fine, but only on the condition that they are preprocessed in a trial-independent way.

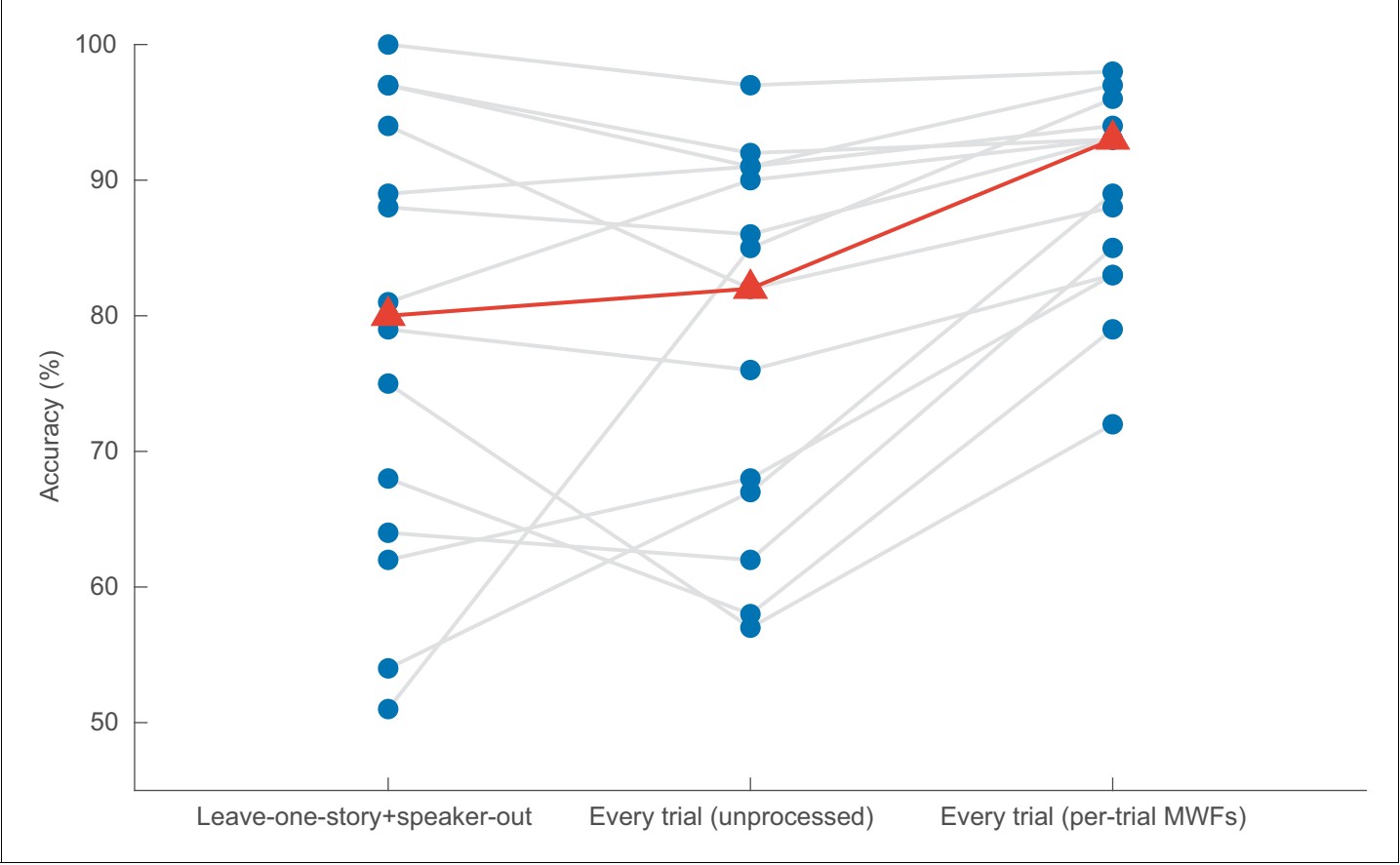

**Figure 7.** Impact of the model validation strategy on the performance of the CNN (decision windows of 1 s). In *Leave-one-story+speaker-out*, the training set does not contain examples of the speakers or stories that appear in the test set. In *Every trial (unprocessed)*, the training, validation, and test sets are extracted from every trial (although always disjoint), and no spatial filtering takes places. In *Every trial (per-trial MWFs)*, data is again extracted from every trial, but this time per-trial MWF filters are applied. CNN = convolutional neural network.

### Subject-independent decoding

In a final experiment, we investigated how well the CNN performs on subjects that were not part of the training set. Here, the CNN is trained on $N-1$ subjects and tested on the held-out subject – but still in a *leave-one-story+speaker* out manner, as before. The results are shown in *Figure 8*. For windows of 1 s, a Wilcoxon signed-rank test shows that leaving out the test subject results in a significant decrease in decoding accuracy from 80.8% to 69.3% ($W = 14$, $p = 0.0134$). Surprisingly, for one subject the network performs better when its data was not included during training. Other window lengths show similar results.

## Discussion

We proposed a novel CNN-based model for decoding the direction of attention (left/right) without access to the stimulus envelopes, and found it to significantly outperform a linear decoder that was trained to reconstruct the envelope of the attended speaker.

### Decoding accuracy

The CNN model resulted in a significant increase in decoding accuracy compared to the linear model: for decision windows as low as 1 s, the CNN's median performance is around 81%. This is also better than entropy-based direction classification presented in literature (*Lu et al., 2018*), in which the average decoding performance proved to be insufficient for real-life use (less than 80%

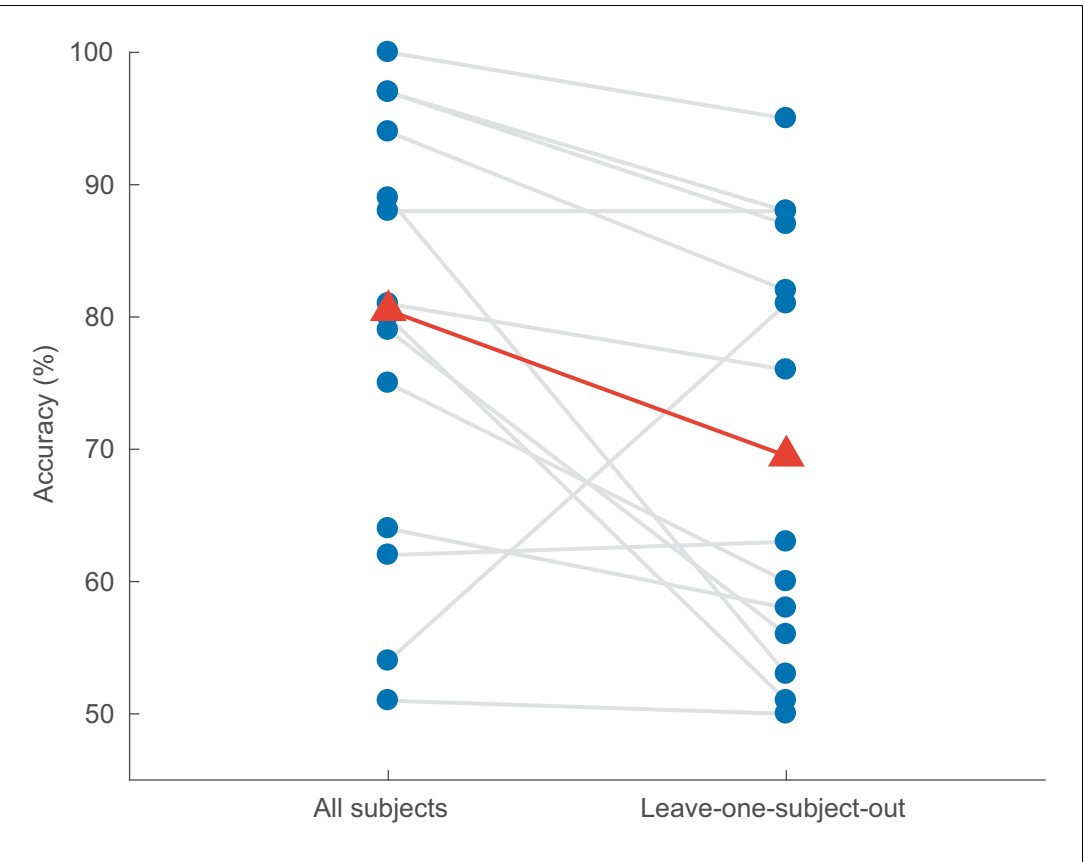

**Figure 8.** Impact of leaving out the test subject on the accuracy of the CNN model (decision windows of 1 s). Blue dots: per-subject results, averaged over two test stories. Gray lines: same subjects. Red triangles: median accuracies. CNN = convolutional neural network.

for decision windows of 60 s). Moreover, our network achieves an unprecedented median MESD of only 0.819 s, compared to 22.6 s for the linear method, allowing for robust neuro-steered volume control with a practically acceptable latency.

Despite the impressive median accuracy of our CNN, there is clearly more variability between subjects in comparison to the linear model. *Figure 4*, for example, shows that some subjects have an accuracy of more than 90%, while others are at chance level – and two subjects even perform below chance level. While this increase in variance could be due to our dataset being too small for the large number of parameters in the CNN, we observed that the poorly performing subjects do better on stories 3 and 4, which were originally excluded as a test set in the cross-validation. Why our system performs poorly on some stories, and why this effect differs from subject to subject, is not clear, but nevertheless it does impact the per-subject results. This story-effect is not present in the linear model, probably because that model has far fewer parameters and is unable to pick up certain intricacies of stories or speakers.

As expected, we found a significant effect of decision window length on accuracy. This effect is, however, clearly different for the two models: the performance of the CNN is much less dependent on window length than is the case for the linear model. For the CNN, going from 10 s to 1 s, the median accuracy decreases by only 4.3% (from 85.1% to 80.8%), while with the linear model it decreases by 17.6% (from 75.7% to 58.1%). Moreover, even at 0.25 s the CNN still achieves a median accuracy of 74.0%, compared to only 53.4% for the linear model. We hypothesize that this difference is because the CNN does not know the stimulus and is only required to decode the locus of attention. As opposed to traditional AAD techniques, it does not have to relate the neural activity to the underlying speech envelopes. The latter requires computing correlation coefficients between

the stimulus and the neural responses, which are only sufficiently reliable and discriminative when computed over long windows.

As usual with deep neural networks, it is hard to pinpoint exactly which information the system uses to achieve attention decoding. Potential information sources are spatial patterns of brain activity related to auditory attention, but also eye gaze or (ear) muscle activity which can be reflected in the EEG. While the subjects most likely focused on a screen in front of them and were instructed to sit still, and we conducted a number of control experiments such as removing the frontal EEG channels, none of these arguments or experiments was fully conclusive, so we can not exclude the possibility that information from other sources than the brain was used to decode attention.

Lastly, we evaluated our system using a *leave-one-story+speaker-out* approach, which is not commonly done in the literature. The usual approach is to leave out a single trial without consideration for speaker and/or story. This is probably fine for linear models, but we wanted to see whether the same would hold for a more complex model such as a CNN. Our results demonstrate that, when properly preprocessing the data, there is no significant difference in decoding accuracy between the *leave-one-story+speaker-out* approach and the classical approach. However, strong overfitting effects were observed when a per-trial (data-driven) preprocessing is performed, for example, for artifact removal. This implies that the data-driven procedure generates intertrial differences in the spatio-temporal data structure that can be exploited by the network. We conclude that one should be careful when applying data-driven preprocessing methods such as independent component analysis, principal component analysis, or MWF in combination with spatio-temporal decoders. In such cases, it is important not to run the preprocessing on a per-trial basis, but run it only once on the entire recording to avoid adding per-trial fingerprints that can be discovered by the network.

## Future improvements

We hypothesize that much of the variation within and across subjects and stories currently observed is due to the small size of the dataset. The network probably needs more examples to learn to generalize better. However, a sufficiently large dataset, one which also allows for the strict cross-validation used in this work, is currently not available.

Partly as a result of the limited amount of data available, the CNN proposed in this work is relatively simple. With more data, more complex CNN architectures would become feasible. Such complex CNN architectures may benefit more from generalization features such as dropout and batch normalization, not discussed in this work.

Also, for a practical neuro-steered hearing aid, it may be beneficial to make soft decisions. Instead of the translation of the continuous softmax outputs into binary decisions, the system could output a probability of left or right being attended, and the corresponding noise suppression system could adapt accordingly. In this way the integrated system could benefit from temporal relations or the knowledge of the current state to predict future states. The CNN could for example be extended by a long short term memory (LSTM) network.

## Applications

The main bottleneck in the implementation of neuro-steered noise suppression in hearing aids thus far has been the detection speed (state-of-the-art algorithms only achieve reasonable accuracies when using long decision windows). This can be quantified through the MESD metric, which captures both the effect of detection speed and decoding accuracy. While our linear baseline model achieves a median MESD of 22.6 s, our CNN achieves a median MESD of only 0.819 s, which is a major step forward.

Moreover, our CNN-based system has an MESD of 5 s or less for 11 out of 16 subjects (eight subjects even have an MESD below 1 s), which is what we assume the minimum for an auditory attention detection system to be feasible in practice. Note that while a latency of 5 s may at first sight still seem long for practical use, it should not be confused with the time it takes to actually *start* steering toward the attended speaker: the user will already hear the effect of switching attention sooner. Instead, the MESD corresponds to the total time it takes to switch an AAD-steered volume control system from one speaker to the other in a *reliable* fashion by introducing an optimized amount of 'inertia' in the volume control system to avoid spurious switches due to false positives (*Geirnaert et al., 2020*). (For reference, an MESD of 5 s corresponds to a decoding accuracy of 70%

at 1 s.) On the other hand, one subject does have an MESD of 33.4 s, and two subjects have an infinitely high MESD due to below 50% performance. The intersubject variability thus remains a challenge, since the goal is to create an algorithm that is both robust and able to quickly decode attention within the assumed limits for all subjects.

Another difficulty in neuro-steered hearing aids is that the clean speech envelopes are not available. This has so far been addressed using sophisticated noise suppression systems (*Van Eyndhoven et al., 2017*; *O'Sullivan et al., 2017*; *Aroudi et al., 2018*). If the speakers are spatially separated, our CNN might elegantly solve this problem by steering a beamformer toward the direction of attention, without requiring access to the envelopes of the speakers at all. Note that in a practical system, the system would need to be extended to more than two possible directions of attention, depending on the desired spatial resolution.

For application in hearing aids, a number of other issues need to be investigated, such as the effect of hearing loss (*Holmes et al., 2017*), acoustic circumstances (e.g., background noise, speaker locations and reverberation [*Das et al., 2018*; *Das et al., 2016 Fuglsang et al., 2017*; *Aroudi et al., 2019*]), mechanisms for switching attention (*Akram et al., 2016*), etc. The computational complexity would also need to be reduced. Especially if deeper, more complex networks are designed, CNN pruning will be necessary (*Anwar et al., 2017*). Then, a hardware DNN implementation or even computation on an external device such as a smartphone could be considered. Another practical obstacle is the numerous electrodes used for the EEG measurements. Similar to the work of *Mirkovic et al., 2015*; *Mundanad and Bertrand, 2018*; *Fiedler et al., 2016*; *Montoya-Martínez et al., 2019*, it should be investigated how many and which electrodes are minimally needed for adequate performance.

In addition to potential use in future hearing devices, fast and accurate detection of the locus of attention can also be an important tool in future fundamental research. Thus far, it was not possible to measure compliance of the subjects with the instruction to direct their attention to one ear. Not only may the proposed CNN approach enable this, but it will also allow to track the locus of attention in almost real-time, which can be useful to study attention in dynamic situations, and its interplay with other elements such as eye gaze, speech intelligibility and cognition.

In conclusion, we proposed a novel EEG-based CNN for decoding the locus of auditory attention (based only on the EEG), and showed that it significantly outperforms a commonly used linear model for decoding the attended speaker. Moreover, we showed that the way the model is trained, and the way the data is preprocessed, impacts the results significantly. Although there are still some practical problems, the proposed model approaches the desired real-time detection performance. Furthermore, as it does not require the clean speech envelopes, this model has potential applications in realistic noise suppression systems for hearing aids.

## Acknowledgements

The work was funded by KU Leuven Special Research Fund C14/16/057 and C24/18/099, Research Foundation Flanders (FWO) project nos. 1.5.123.16N and G0A4918N, the European Research Council (ERC) under the European Union's Horizon 2020 research and innovation programme (grants no. 637424 [T Francart] and no. 802895 [A Bertrand]) and the Flemish Government under the 'Onderzoeksprogramma Artificiele Intelligentie (AI) Vlaanderen' program. A Ansari is a postdoctoral fellow of the Research Foundation Flanders (FWO). We thank Simon Geirnaert for his constructive criticism and for help with some of the technical issues we encountered.

## Additional information

### Funding

| Funder | Grant reference number | Author |
| --- | --- | --- |
| KU Leuven | C14/16/057 | Tom Francart |
| KU Leuven | C24/18/099 | Alexander Bertrand |
| Research Foundation Flanders | 1.5.123.16N | Alexander Bertrand |
| Research Foundation Flanders | G0A4918N | Alexander Bertrand |

| European Research Council | 637424 | Tom Francart |
| European Research Council | 802895 | Alexander Bertrand |

The funders had no role in study design, data collection and interpretation, or the decision to submit the work for publication.

## Author contributions

Servaas Vandecappelle, Conceptualization, Resources, Data curation, Software, Formal analysis, Validation, Investigation, Visualization, Methodology, Writing - original draft, Writing - review and editing; Lucas Deckers, Conceptualization, Resources, Software, Formal analysis, Validation, Investigation, Visualization, Methodology, Writing - original draft; Neetha Das, Conceptualization, Resources, Data curation, Writing - review and editing; Amir Hossein Ansari, Conceptualization, Software, Supervision, Validation, Writing - review and editing; Alexander Bertrand, Tom Francart, Conceptualization, Resources, Supervision, Funding acquisition, Validation, Methodology, Project administration, Writing - review and editing

## Author ORCIDs

Servaas Vandecappelle https://orcid.org/0000-0002-0266-7293
Alexander Bertrand http://orcid.org/0000-0002-4827-8568

## Ethics

Human subjects: The experiment was approved by the Ethics Committee Research UZ/KU Leuven (S57102) and every participant signed an informed consent form approved by the same commitee.

## Decision letter and Author response

Decision letter https://doi.org/10.7554/eLife.56481.sa1
Author response https://doi.org/10.7554/eLife.56481.sa2

# Additional files

## Supplementary files

- Transparent reporting form

## Data availability

Code used for training and evaluating the network has been made available at https://github.com/exporl/locus-of-auditory-attention-cnn (copy archived at https://archive.softwareheritage.org/swh:1:rev:3e5e21a7e6072182e076f9863ebc82b85e7a01b1). The CNN models used to generate the results shown in the paper are also available at that location. The dataset used in this study had been made available earlier at https://zenodo.org/record/3377911.

The following previously published dataset was used:

| Author(s) | Year | Dataset title | Dataset URL | Database and Identifier |
|---|---|---|---|---|
| Vandecappelle S, Deckers L,  Das N, Ansari AH,  Bertrand A,  Francart T | 2019 | Auditory Attention Detection Dataset KULeuven | https://doi.org/10.5281/zenodo.3377911 | Zenodo, 10.5281/zenodo.3377911 |

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
