## [Decision Letter]

**Acceptance summary:**

This paper aims to assess how well attention to a speaker can be decoded from EEG using convolutional neural networks (CNNs). In particular, the authors train a CNN on EEG data from a "cocktail party" attention experiment and demonstrate impressive decoding performance, better than many prior related efforts. Though effects of eye gaze cannot be completely ruled out, the authors acknowledge this potential confound and do a diligent job of addressing this concern. These provocative results are likely to impact future research in the use of EEG to decode the focus of attention in auditory tasks.

**Decision letter after peer review:**

[Editors’ note: the authors submitted for reconsideration following the decision after peer review. What follows is the decision letter after the first round of review.]

Thank you for submitting your work entitled "EEG-based detection of the attended speaker and the locus of auditory attention with convolutional neural networks" for consideration by *eLife*. Your article has been reviewed by 3 peer reviewers, including Barbara G Shinn-Cunningham as the Reviewing Editor and Reviewer #1, and the evaluation has been overseen by a Senior Editor. The following individual involved in review of your submission has agreed to reveal their identity: Andrew Dimitrijevic (Reviewer #2).

Our decision has been reached after consultation between the reviewers. Based on these discussions and the individual reviews below, we regret to inform you that your work will not be considered further for publication in *eLife*.

All of the reviewers felt that the work has the potential to appear in *eLife*. However, there were substantial concerns about some of the technical details. Without some significant additional work to address potential limitations of the findings and confounds of the experiments that were conducted, we felt the manuscript was not ready for publication in *eLife*. The standard for asking for a revision (rather than a rejection) for *eLife* is that if any additional work is likely to take two months or more. Given this, we must reject the manuscript: we believe that the additional work required will take more than two months.

*Reviewer #1:*

This is an interesting paper that addresses a very timely and interesting question. Given the attention (pun intended) to real-time decoding of attention in the field today, the approach described is likely to be influential.

However, as written, I am not sure how general the findings are, based on the experiments described. Reviewer 3 does an excellent job of articulating the concerns I had, as well, so I am not reiterating them here. With additional controls that demonstrate the robustness of the findings, this work will be of high impact.

Overall, the paper is very clearly written. However, there are a few phrasings that are grammatically proper, but that sound awkward to a native English speaker's ear. (For instance, "Especially the elderly and people suffering from hearing loss have difficulties attending to one person in a noisy environment." is more natural when written as "Both the elderly and people suffering from hearing loss have particular difficulty attending to one person in a noisy environment." ) If the paper were being revised at this point, I would offer a more complete list of such sentences and suggested edits-- but don't believe it makes sense to do so at this juncture.

*Reviewer #2:*

The manuscript "EEG-based detection of the attended speaker and the locus of auditory attention with convolutional neural networks" describes a study where the authors used a convolutional neural network (CNN) to identify auditory attend locations while the EEG was recorded. The data indicated that CNNs can classify attend locations and accuracy and speed of detection increases when the stimulus envelope is included in the CNN.

As written, the manuscript may appeal to engineering or computer science audiences, however, I feel that more needs to be incorporated to appeal to a broader scope/readership of *eLife*. Although this may deter from the practical or real-world application of the CNN, including and relating more physiological/neuroscience aspects of the CNN may make the manuscript more palatable to a general audience. It may also demonstrate that this technique can also be used to inform how the brain operates. Two current theories relating to auditory selective attention, as the authors mention, is enhancement of envelope encoding schemes and α lateralization. What features is the CNN using? The use of filtered EEG (low frequency for envelope and band-pass 8-12 Hz, for α) may provide some indication. Some detail on the inferred neural generators, perhaps a topography of the feature weights (similar to de Taillez) would be informative. Also, more detail on the filters used for the spatial-temporal feature map would be helpful. The authors may also consider using a "control" condition to estimate false positive rates. This might be implemented as random EEG shuffling (left and right) for the final testing phase, which would have an accuracy of 50%. Some discussion on the behavioral aspect of the subject performance would also be desirable. Where there content questions about the attend speakers, did the subjects indeed listen to the appropriate target? In cases where CNN performance was not 100% was the subject "peaking a listen" to the other side?

Overall, the CCN is a novel application in this domain and determining attend location within 1-2 sec is a remarkable feat.

*Reviewer #3:*

This is a very nice study, and well written. The applications are very relevant, and the work is timely. However, I have a number of concerns which need to be addressed before I can believe these very impressive results.

The classification performance for the CNN:D model is very high, with accuracy using 1 second of data almost as high as that at 10 seconds. One potential downfall of CNNs (and DNNs in general) is that they might be hyper-sensitive to the particular EEG setup that they're trained on. I.e., if you tested the same subject on another day, would the performance be the same? Or are they learning to optimize performance with a particular setup of electrode locations and noise conditions? I understand that the data set was collected a few years ago, but is it possible to run the experiment again on a small subset of subjects, and use the CNN that was trained on the previous experiment to classify the data from the new experiment? This would address the concern of the CNN overfitting to the precise experimental setup of the day.

The benefit of the linear stimulus reconstruction approach is that we know how it works, and it can generalize to unseen speakers. The authors state that they tried training a DNN to perform stimulus reconstruction, but its performance was not as impressive as the CNN:S+D approach. However, the CNN:S+D specifically requires a binary decision between 2 speakers. Is it possible that the network is over-fitting to the specific speakers in the training set? If 2 new speakers were introduced, could it handle that? Is it possible for the authors to test this with the current data-set? If not, an additional experiment would be required.

In addition, the linear stimulus reconstruction approach allows for a generic subject-independent model that can decode the attention of an unseen subject. The authors do show results from a generic CNN, but this was trained on all subjects. Can the authors perform an additional analysis using a generic decoder but ensure that the test subject has been completely unseen by the network?

On a similar note, the training, cross-validation, and test data were all obtained from the same trials. I.e., in a single 6 minute trial, the first part was chosen as the training set, followed by cross-validation and test sets. This could lead to overly optimistic results. Can the authors perform an additional analysis where the training, validation, and test sets are all taken from different trials?

Can the authors provide any insight into what the network is learning, and how it can perform so well? As the authors mention in the introduction, perhaps it is α power. They could test this hypothesis by providing the CNN with different frequency bands of the neural data.

In summary, I would require to see a lot more proof that the CNN is not just overfitting to the particular subject, EEG setup, and day of recording, and that these results are generalizable.

[Editors’ note: further revisions were suggested prior to acceptance, as described below.]

Thank you for submitting your article "EEG-based detection of the locus of auditory attention with convolutional neural networks" for consideration by *eLife*. Your article has been reviewed by 3 peer reviewers, and the evaluation has been overseen by Barbara Shinn-Cunningham as the Senior and Reviewing Editor. The following individuals involved in review of your submission have agreed to reveal their identity: James O'Sullivan (Reviewer #1); Andrew Dimitrijevic (Reviewer #2).

The reviewers have discussed the reviews with one another and the Reviewing Editor has drafted this decision to help you prepare a revised submission.

Summary:

This manuscript presents research aimed at assessing how well attention to a speaker can be decoded from EEG using convolutional neural networks. In particular, the authors train a convolutional neural network directly on EEG data during a "cocktail party" attention experiment and compare it to an approach based on based on reconstructing an estimate of the speech envelope from the EEG using linear regression. The authors demonstrate decoding performance n with accuracies of ~80% using just 1-2 s of data, which is much better than the state of the art.

The reviewers all believe that this work may be appropriate for a Tools and Methods paper in *eLife*. However, there remain a few critical questions and concerns that need to be addressed for the paper to make its contribution to the field clear.

There are some potential strengths of this technical report comparing the CNNs and linear models for decoding auditory spatial attention using EEG. This research opens new avenues of exploration of auditory attention methods that can be used for real-time decoding applications such as neurally steered hearing aids. The authors claim that it is possible to decode the locus of attention with accuracies of ~80% using just 1-2 s of data is much better than the state of the art.

Because we could not obtain assessments from all of the original reviewers, one of the reviewers is new to the paper. This reviewer read the paper and wrote their own comments before going back and looking at the earlier reviews. They noted that some of the points that concerned them had been raised before. Still, the reviewers who saw your earlier submission do appreciate the changes you made.

The remaining critical issues that must be addressed for the paper to be published are:

1. Comparing current results to those obtained using envelope reconstruction is useful, but it is somewhat unfair. That is something that you should acknowledge. Specifically, the envelope reconstruction approach is not just a linear approach, it is a linear approach that is constrained to relating EEG responses to the envelopes of the two speech streams. No such constraint is placed on the CNN; it trains on the EEG and settles on whatever features are best for solving the question. Related to this, even the EEG preprocessing (filtering) is different for the CNN and the envelope reconstruction approaches. While this makes sense (the filters chosen for the envelope reconstruction seemed reasonable based on the literature), it also means that the information in the EEG differs in the two analyses. These issues should be acknowledged.

2. Some explanations of what features drive the CNN performance would greatly increase the impact of the paper. As a Tools and Methods paper, there are not significant expectations for demonstration of important neuroscience findings. Still, without some information about what is happening in the neural responses, readers cannot judge the likely usefulness and replicability of this "tool." Is there any way to know this? For example, some of the cited literature (e.g., Bednar; Wostmann) show that α power is important for decoding spatial attention. Α frequencies are included in your CNN analysis and might be responsible for the results you describe. You could check this by seeing how the CNN performance drops if you exclude α frequencies, for instance.

Relatedly, it is almost worrying how good the performance gets when you train on the other examples from the same story and speaker (Figure 5). Why would this be? Is the CNN picking up on some weird features in the EEG that are very specific to these speakers? Without having a sense of what drives the exceptional performance, it makes one wonder what the CNN relies on.

3. The results presented in the manuscript show no effect of window size on performance. This must, in the limit, not be true. More data must be shown to show this dependence and determine the limits of the method.

4. For 3 subjects, with a 10s window, the performance of the CNN was lower than the linear model (Figure 2). How is it possible then, that every subject had a better MESD when using the CNN (Figure 3)? I know you've excluded 1 subject from the figure, but what about the other 2 subjects?

5. You talk about the idea that future work can address some unanswered questions, like whether or not performance will drop with fewer EEG channels. However, related to the idea that the results might be driven be decoding of spatial attention, it would be interesting to know if spatial patterns are driving the CNN decoding.

[Editors' note: further revisions were suggested prior to acceptance, as described below.]

Thank you for resubmitting your article "EEG-based detection of the locus of auditory attention with convolutional neural networks" for consideration by *eLife*. Your revised article has been reviewed by 3 peer reviewers, and the evaluation has been overseen by Barbara Shinn-Cunningham as the Senior and Reviewing Editor. The following individuals involved in review of your submission have agreed to reveal their identity: Andrew Dimitrijevic (Reviewer #2); Behtash Babadi (Reviewer #4).

The reviewers have discussed the reviews with one another and the Reviewing Editor has drafted this decision to help you prepare a revised submission.

Summary

This is a very interesting and provocative paper, which demonstrates decoding from EEG of the directional focus of auditory attention in a dichotic or HTRF-emulated competing-speaker setting. Using a CNN-based decoder to jointly extract the relevant features and classify the locus of attention, you show significant decoding improvements compared to the common linear decoding techniques; moreover, the decoding is rapid, and is thus able to track attentional switches. Analyses implicate the β band as well as frontal EEG channels in decoding. The paper is well-written and clear, the methods are described carefully and transparently, the results are impressive, and the discussion is thorough and inspiring. Your cross-validation scheme for training the CNN to avoid overfitting is admirable; this is very often overlooked.

In addition, we would like to note how thoughtfully you revised your original submission. Two of the three original reviewers read this revision, along with one new reviewer. It was clear to all that your revision and your reply to the previous criticisms were responsive and thorough. We want to thank and commend you for the work you put in on this revision. That said, the revision raised a new concern, discussed below.

Essential revision

1. The reviewers are not convinced that eye movements are not a substantial contributor to decoding accuracy. Specifically, the frontal topography of the convolution filters in Figure 6 looks suspiciously like an EOG signature. We think it is critical for you to clarify what features of the EEG are being used for classification. One way to test this would be look at the raw data (attend left vs right) and look the time-frequency profile.

1a. Saccade-related ERP profiles tend to have a positive peak near 0 ms followed by a negative peak around 20 ms. The attention-related ERPs using EEG, however, have key peaks at in the 100-200 ms range. Given this, the temporal profile of the filters may inform the arguments for and against eye movements contributing.

1b. Relatedly, if you found that the filters were tuned to γ band activity, this would suggest that small saccades are influencing performance. The fact that the network weights the β band as much as it does suggests that it may even like γ band more. On the other hand, if the filters are tuned to α or high δ, that would argue against saccades being the cause.

1c. Your MWF algorithm should remove large gaze artifacts. However, even very small (but consistent) gaze changes could be responsible for some of the effects you see. You should also consider the literature on micro saccades and γ, and about whether small but consistent drifts of gaze during long trials contribute.

1d. We are aware of your recent arXiv paper (Geirnaert et al) in which the CNN fails on another data set. Were subjects asked to fixate in that study, but not this? A better description of how subjects were instructed in the current study should be included, no matter what. Given the Geirnaert results, we think it is especially critical to figure out whether the results in the current paper really are attention effects in neural responses, rather than due to eye movement. It would be unfortunate to have to publish a correction if the results in the current study are attributed to attentional effects when they are actually due to gaze differences.

Given these issues, we would like you to undertake some of the above analyses to address the concerns, and consider in the Discussion the evidence for and against eye gaze contributing to the exceptional performance of your algorithm.

---

## [Author Response]

[Editors’ note: the authors resubmitted a revised version of the paper for consideration. What follows is the authors’ response to the first round of review.]

Reviewer #1:(1) This is an interesting paper that addresses a very timely and interesting question. Given the attention (pun intended) to real-time decoding of attention in the field today, the approach described is likely to be influential.However, as written, I am not sure how general the findings are, based on the experiments described. Reviewer 3 does an excellent job of articulating the concerns I had, as well, so I am not reiterating them here. With additional controls that demonstrate the robustness of the findings, this work will be of high impact.

We understand the concerns that Reviewer 1 and Reviewer 3 have regarding the robustness of our findings. We have made extensive changes to our experimental paradigm to address this. Because it was mostly Reviewer 3 who articulated the concerns, our answers are given below in Reviewer 3’s section. In this section we limit ourselves to additional comments made by Reviewer 1.

(2.1) Overall, the paper is very clearly written. However, there are a few phrasings that are grammatically proper, but that sound awkward to a native English speaker's ear. (For instance, "Especially the elderly and people suffering from hearing loss have difficulties attending to one person in a noisy environment." is more natural when written as "Both the elderly and people suffering from hearing loss have particular difficulty attending to one person in a noisy environment." )

We very much welcome and appreciate comments regarding the readability of our paper; we confess that we are not native speakers of English. We have revised the language to the best of our ability, but admit that improvement is undoubtedly still possible.

(2.2) If the paper were being revised at this point, I would offer a more complete list of such sentences and suggested edits but don't believe it makes sense to do so at this juncture.

Thank you, we look forward to any further comments you may have.

Reviewer #2:(1) The manuscript "EEG-based detection of the attended speaker and the locus of auditory attention with convolutional neural networks" describes a study where the authors used a convolutional neural network (CNN) to identify auditory attend locations while the EEG was recorded. The data indicated that CNNs can classify attend locations and accuracy and speed of detection increases when the stimulus envelope is included in the CNN.

We thank the reviewer for the positive remark. To avoid misunderstanding, we note that with the adjustments made to the way the network is trained (which we elaborately explain in our comments to Reviewer 3), there was no longer a significant statistical difference between the CNN that incorporates envelopes (previously called “CNN:S+D”) and the CNN that does not (previously called “CNN:D”). It is for that reason that we decided to no longer include CNN:S+D and instead focus on CNN:D. Nevertheless, the speed of detection of this CNN:D network is a major step forward compared to the reported detection times in the recent literature.

(2.1) As written, the manuscript may appeal to engineering or computer science audiences, however, I feel that more needs to be incorporated to appeal to a broader scope/readership of eLife. Although this may deter from the practical or real-world application of the CNN, including and relating more physiological/neuroscience aspects of the CNN may make the manuscript more palatable to a general audience. It may also demonstrate that this technique can also be used to inform how the brain operates. Two current theories relating to auditory selective attention, as the authors mention, is enhancement of envelope encoding schemes and α lateralization. What features is the CNN using? The use of filtered EEG (low frequency for envelope and band-pass 8-12 Hz, for α) may provide some indication. Some detail on the inferred neural generators, perhaps a topography of the feature weights (similar to de Taillez) would be informative. Also, more detail on the filters used for the spatial-temporal feature map would be helpful.

We agree with the reviewer that insight into how the network operates, and what it learns exactly, would be informative both for the further development of neural network-based decoders and for neuroscience in general. We have done some elementary analysis to try to have a rough idea of what the network actually does by investigating the spatial and spectral topology of the convolution kernels, but we could not find clear trends. We feel that adding more advanced analyses to try to further open up the black box is beyond the scope and would also lead to an overloaded paper (both in terms of methodology and results). We also respectfully point out that the manuscript was submitted to the Tools and Resources category, for which the author guide states “This category highlights tools or resources that are especially important for their respective fields and have the potential to accelerate discovery. […] Tools and Resources articles do not have to report major new biological insights or mechanisms”. Though very interesting, we would prefer to keep the paper crisp and stick to the analysis of the performance of the network.

(3) The authors may also consider using a "control" condition to estimate false positive rates. This might be implemented as random EEG shuffling (left and right) for the final testing phase, which would have an accuracy of 50%.

Thank you for the comment. We agree that a control condition is useful. However, given that the D network (which is now the exclusive focus of the new version of the manuscript) only uses EEG and no stimulus information, each EEG segment has a “correct answer”, so shuffling the EEG in time relative to the stimulus would not work.

(4) Some discussion on the behavioral aspect of the subject performance would also be desirable. Where there content questions about the attend speakers, did the subjects indeed listen to the appropriate target? In cases where CNN performance was not 100% was the subject "peaking a listen" to the other side?

In the Das et al. 2016 dataset (which we use in this study), attention was measured behaviorally with a multiple-choice quiz given after every 6 min trial. We recognize this was not clearly explained in the manuscript and have added a more extensive description of the experimental setup. Note that there was no significant correlation found between performance on this quiz and the attention decoding performance.

Reviewer #3:(1) This is a very nice study, and well written. The applications are very relevant, and the work is timely. However, I have a number of concerns which need to be addressed before I can believe these very impressive results.The classification performance for the CNN:D model is very high, with accuracy using 1 second of data almost as high as that at 10 seconds. One potential downfall of CNNs (and DNNs in general) is that they might be hyper-sensitive to the particular EEG setup that they're trained on. I.e., if you tested the same subject on another day, would the performance be the same? Or are they learning to optimize performance with a particular setup of electrode locations and noise conditions? I understand that the data set was collected a few years ago, but is it possible to run the experiment again on a small subset of subjects, and use the CNN that was trained on the previous experiment to classify the data from the new experiment? This would address the concern of the CNN overfitting to the precise experimental setup of the day.

Reviewer 3 makes very relevant comments about the generalization of our findings. We understand why this is a cause of concern and have since made changes to our paradigm to improve the robustness of our results. Below, we reiterate the points above and explain how we addressed them.

Before doing so, however, we would like to point out that the manner with which we trained our model was not unusual. To reiterate, we partitioned each trial (6 minutes of the same story, attended to by the same ear) into a training, validation and testing set. The CNN was therefore never tested on data it had already seen, but one could argue that having already seen a different part of the EEG elicited by the *same story* could lead the model to gain an unfair advantage. The same argument holds for the narrator. As far as we know, this dependency has never been taking into account in other peer-reviewed AAD algorithm papers, though we admit this is probably much less an issue for linear models than for non-linear models.

On the other hand, a recent peer-reviewed paper also proposes a non-linear model (Ciccarelli et al., 2019), and they admit a similar training scheme— though they do not partition individual parts, but instead use a leave-one-trial-out scheme.

Nonetheless, we agree that this is a cause of concern and we have made steps to eliminate this potential dependency.

(2.1) i.e., if you tested the same subject on another day, would the performance be the same? […] I understand that the data set was collected a few years ago, but is it possible to run the experiment again on a small subset of subjects, and use the CNN that was trained on the previous experiment to classify the data from the new experiment? This would address the concern of the CNN overfitting to the precise experimental setup of the day.

This is an excellent point. Although we are also curious as to how this would impact the model performance, we regret to say we are unable to repeat the experiment. Our main results are based on a subject-dependent model that requires example data of the test subject—and unfortunately we can no longer retest the same subjects. Due to a large time gap we were unable to recruit the same students from the Das et al. (2016) study to do a re-test, as they have since moved on. Note that, as opposed to the previous version of the manuscript, the network is now trained on data from all subjects which in itself acts as a regularizer to avoid that the network overfits to one particular experiment on one particular day. The subject/experiment-dependent post-training has now been omitted as it was observed to not improve performance. If the network were to benefit from learning experiment/day-specific features, an improvement would be expected here, which is not the case.

(2.2) Or are they learning to optimize performance with a particular setup of electrode locations and noise conditions?

Properly testing these confounds would entail repeating the experiment on the same subject. We refer to (2.1) for our rationale as to why we are regrettably unable to do so. Note that we do expect that the network is indeed dependent on the electrode locations, as the initial convolutional layer is a spatial filter, which must deteriorate if the test data and train data would use different locations on the scalp. However, the same holds for all multi-channel (backwards) decoders in the current literature on AAD. We believe it is reasonable to assume a pre-fixed montage.

(2.3) The benefit of the linear stimulus reconstruction approach is that we know how it works, and it can generalize to unseen speakers. The authors state that they tried training a DNN to perform stimulus reconstruction, but its performance was not as impressive as the CNN:S+D approach. However, the CNN:S+D specifically requires a binary decision between 2 speakers. Is it possible that the network is over-fitting to the specific speakers in the training set? If 2 new speakers were introduced, could it handle that? Is it possible for the authors to test this with the current data-set? If not, an additional experiment would be required.

To be clear, we had four stories in total; two were narrated by two different speakers, and two by the same speaker. In our original train/val/test setup, both the train and test sets contained EEG elicited by speech of the same speaker (although never the same part of the story). We agree that having already seen (EEG elicited by) the same speaker could lead to overly optimistic model performance. We have therefore made changes so that the model is trained in a *leave-one-story+speaker-out* way. That means:

1. For *leave-one-story-out*: Per subject, we partitioned the data into four subsets, one for each (attended) story. During training we then iterated over the four stories, taking the current story as the test story, while the other three were used for training. That way the network was tested on a story it had never seen before. The performance was defined as the average performance over the four folds.

**Author response table 1. resptable1:** Leave-one-story-out scheme. Example of one out of four folds. In this particular fold, the test set consists of story 1, and the training and validation sets consist of stories 2, 3, and 4. Training and validation sets are completely separate from the test set. Per-subject accuracies are based on a subject-specific test set (noted by multiple mentions of "test" in Author response table 1). The model is trained on data of all subjects (noted by a single mention of "train/val").

Story	Subject 1	Subject 2	….	Subject 16
1	test	test	test	test
2	train/val			
3	train/val			
4	train/val			

2. For *leave-one-speaker-out*, we note that story 3 and 4 were narrated by the same speaker. As a consequence, in two of the four folds, the test story and one of the three training stories were narrated by the same speaker. To also exclude the effects of speaker dependency, we discarded those two folds. The performance was then defined as the average of the two other folds. (This had no consequences on the amount of training data—in each fold the network was still trained on three stories and tested on one.)

We decided on this combined *speaker+story* scheme because it was the only way to eliminate both confounds without having to collect new data.

(Note that the Das et al. (2016) dataset is balanced in terms of attended ear—also on the level of stories—and that, hence, the folds are also balanced in that regard.)

(2.4) In addition, the linear stimulus reconstruction approach allows for a generic subject-independent model that can decode the attention of an unseen subject. The authors do show results from a generic CNN, but this was trained on all subjects. Can the authors perform an additional analysis using a generic decoder but ensure that the test subject has been completely unseen by the network?

Thank you for the suggestion. Note that the generic CNN mentioned by the reviewer is in fact a subject-specific decoder, as training data from the subject under test was included (yet other subjects were included in the training set to increase the training data and avoid overfitting). To avoid confusion with a subject-independent decoder, we avoid the term “generic” to describe such a decoder.

As suggested by the reviewer, we have added a section where we show results of a model trained on *N*− 1 subjects and tested on the unseen subject. We show that there is a significant drop in median accuracy, but that the decoding accuracy remains above 70% for 7 out of 16 subjects. We feel that this is an additional strength of our model, and certainly something we would like to further explore in the future.

(3) On a similar note, the training, cross-validation, and test data were all obtained from the same trials. I.e., in a single 6 minute trial, the first part was chosen as the training set, followed by cross-validation and test sets. This could lead to overly optimistic results. Can the authors perform an additional analysis where the training, validation, and test sets are all taken from different trials?

Thank you for pointing this out. This was mainly answered in our response to (2.3), but we would like to point out again that while our test set now comes from held-out stories/speakers, our training and validation are still taken from the same trials. This should result in a model that does well on the “average” of the three training stories, rather than on one particular story (which would be the case when the validation set consists of one story only). This does not cause an issue with generalizability or over-optimistic test results, however, because the test story and speaker are still completely unseen. In that sense we feel we have satisfied the reviewer’s request.

(4) Can the authors provide any insight into what the network is learning, and how it can perform so well? As the authors mention in the introduction, perhaps it is α power. They could test this hypothesis by providing the CNN with different frequency bands of the neural data.

We certainly acknowledge this would be interesting, but for reasons explained in Reviewer 2’s section, we would rather not extend the scope at this time.

[Editors’ note: what follows is the authors’ response to the second round of review.]

Revisions for this paper:The remaining critical issues that must be addressed for the paper to be published are:1. Comparing current results to those obtained using envelope reconstruction is useful, but it is somewhat unfair. That is something that you should acknowledge. Specifically, the envelope reconstruction approach is not just a linear approach, it is a linear approach that is constrained to relating EEG responses to the envelopes of the two speech streams. No such constraint is placed on the CNN; it trains on the EEG and settles on whatever features are best for solving the question. Related to this, even the EEG preprocessing (filtering) is different for the CNN and the envelope reconstruction approaches. While this makes sense (the filters chosen for the envelope reconstruction seemed reasonable based on the literature), it also means that the information in the EEG differs in the two analyses. These issues should be acknowledged.

Thank you for the insightful comment. We agree that the comparison is not obvious and that reader should fully understand the assumptions that are being made. We have added the following paragraph at the end of Section II.E to make this more clear:

“Note that the results of the linear model here merely serve as a representative baseline, and that a comparison between the two models should be treated with care—in part because the CNN is non-linear, but also because the linear model is only able to relate the EEG to the envelopes of the recorded audio, while the CNN is free to extract any feature it finds optimal (though only from the EEG, as no audio is given to the CNN). Additionally, the prepossessing is slightly different for both models. However, that preprocessing was chosen such that each model would perform optimally—using the same preprocessing would in fact negatively impact one of the two models.”

2. Some explanations of what features drive the CNN performance would greatly increase the impact of the paper. As a Tools and Methods paper, there are not significant expectations for demonstration of important neuroscience findings. Still, without some information about what is happening in the neural responses, readers cannot judge the likely usefulness and replicability of this "tool." Is there any way to know this? For example, some of the cited literature (e.g., Bednar; Wostmann) show that α power is important for decoding spatial attention. Α frequencies are included in your CNN analysis and might be responsible for the results you describe. You could check this by seeing how the CNN performance drops if you exclude α frequencies, for instance.

We very much agree with the reviewer that an analysis of how the network works would make the paper more impactful. We do feel that in order to answer this question in full, a thorough and non-trivial analysis is required, one that we certainly want to do in the future but is out of scope for this

particular paper.

We acknowledge that an experiment such as the one suggested by the reviewer could already provide some insight. To that end, we have performed two experiments, one per the suggestion and one variation thereupon:

1. For each frequency band (δ, θ, α, β, with ranges taken from the literature.) X:

– (Original) We filtered the original data such that all frequency bands except X were present.

– (Variation) We filtered the original data such that only X was present.

2. We loaded the original models and evaluated them again on the filtered data.

We have added the results to the "Interpretation of the results" section. In short, we found that our network primarily uses the β-band, rather than the α-band. There is literature that also reports on the importance of the β-band in spatial decoding (i.e., Gao et al. (2017)), which is now also discussed in the paper.

Relatedly, it is almost worrying how good the performance gets when you train on the other examples from the same story and speaker (Figure 5). Why would this be? Is the CNN picking up on some weird features in the EEG that are very specific to these speakers? Without having a sense of what drives the exceptional performance, it makes one wonder what the CNN relies on.

Thank you for the comment. It is indeed true that our experiments show that providing EEG data of the same story and speaker provides a significant and unrealistic benefit, given that in a real-life situation we want our models to generalize to unknown stories and speakers. Previously, we would have made sure to simply take apart testing and training data (as is done in other literature in our field), but clearly this does not suffice and knowing the characteristics of the speaker and/or story in advance helps.

Exactly what drives this is not entirely clear to us. The original experiment (Das et al. 2016) was not designed to investigate this, and due to the way it was set up we feel we can at best only establish this fact. For starters there were only 3 speakers, all male, and there were only 4 stories. As mentioned in the paper, that resulted in only two unique speaker/story combinations. We could include the other combinations, but then it is not clear what the interaction between story and speaker is. To properly investigate this we would expect to do an experiment with many short stories, each narrated by a different speaker.

3. The results presented in the manuscript show no effect of window size on performance. This must, in the limit, not be true. More data must be shown to show this dependence and determine the limits of the method.

Thank you for the comment. We agree that this is worthwhile to have in the paper and have retrained the model on the following window sizes: 0.5s, 0.25s and 0.13s. We choose 0.13s as lowest value since the CNN kernel is also 0.13s wide, which puts a lower bound on the size of the decision window. Also, for the same reason the linear model has not been rerun at 0.13s, since the kernel width there is 0.25s.

We have added the new results to Figure 4, in the "Interpretation of the results" section. The statistical analysis has also been rerun to take into account the new window sizes, and now does show a significant effect of decision window length on performance for the CNN (previously: no effect). The previous result for the linear model remained unchanged.

As a consequence of adding these extra window sizes, the MESD values in the paper have changed slightly.

4. For 3 subjects, with a 10s window, the performance of the CNN was lower than the linear model (Figure 2). How is it possible then, that every subject had a better MESD when using the CNN (Figure 3)? I know you've excluded 1 subject from the figure, but what about the other 2 subjects?

We are grateful to the reviewer for checking our work in such detail. However, this is not an inconsistency. The MESD takes into account all window sizes (initially 10s, 5s, 2s and 1s, but now also even shorter windows), but Figure 2, on the other hand, shows only the results for 10s and 1s. (2s and 5s were not shown because the results were similar to either 10s or 1s.) The particular subjects that the reviewer refers to did indeed have worse results for the CNN than for the linear model, but that was only true for 10s; it was actually the other way around for 1s. And because the MESD metric places more importance on smaller window sizes, those subjects had the best MESD value on the CNN.

5. You talk about the idea that future work can address some unanswered questions, like whether or not performance will drop with fewer EEG channels. However, related to the idea that the results might be driven be decoding of spatial attention, it would be interesting to know if spatial patterns are driving the CNN decoding.

Again an excellent point. We think that spatial patterns must be involved due to the very short window lengths and the fact that no temporal information about the stimulus (envelope) is given as an input to the network. However, similarly to remark (2), it is hard to open the black box and fully understand what spatial patterns the CNN uses. In an attempt to answer this question, we investigated the weights of the convolutional filters by computing a grand-average topographic map, as follows:

1. Calculated the power of each channel in the training set. Normalized exactly as was done during training. This resulted in 64 values, one for each EEG channel.

2. Per model and per filter, calculated the power of the filter coefficients in each channel. Normalized those values by multiplying with the power in the training set, calculated in the previous step. Applied the sqrt to those values, to account for the fact that we want to show power.

3. Performed the above step for each model and each filter, and averaged the results. Normalized the values to lie in the interval [0, 1].

The resulting figure was added to the paper in the "Interpretation of the results" section.

We see primarily activations in the frontal and temporal channels, plus some smaller activations in the occipital lobe. Although that still does not provide us with concrete information regarding the inner workings of the network, it is somewhat in line with other studies in the literature. Ciccarelli et al. (2019), for example, included similar heatmaps of the weights, and also demonstrates strong activity on frontal and temporal channels (although this network also had access to the speech envelope). However, Ciccarelli does not provide any discussion as to what may cause those activations. Additionally, Gao et al. (2017) also found the frontal channels to significantly differ from the other channels within the β band (Figure 3 and Table 1 in Gao et al. (2017)). The prior MWF artefact removal step in the EEG preprocessing and the importance of the β band in the decision making (Figure 5 in the paper) implies that the focus on the frontal channel is not attributed to eye artifacts. It is noted that the filters of the network act as backward decoders, and therefore care should be taken when interpreting topoplots related to the decoder coefficients. As opposed to a forward (encoding) model, the coefficients of a backward (decoding) model are not necessarily predictive for the strength of the neural response in these channels. For example, the network may perform an implicit noise reduction transformation, thereby involving channels with low SNR as well.

[Editors' note: further revisions were suggested prior to acceptance, as described below.]

Essential revision1. The reviewers are not convinced that eye movements are not a substantial contributor to decoding accuracy. Specifically, the frontal topography of the convolution filters in Figure 6 looks suspiciously like an EOG signature. We think it is critical for you to clarify what features of the EEG are being used for classification. One way to test this would be look at the raw data (attend left vs right) and look the time-frequency profile.

While frontal topographies have also been found in other AAD papers (see our Discussion section for a comparison), we wholeheartedly agree that this is an important point and that it might indicate that the network (partially) uses EOG information.

We had a thorough look at the raw data as suggested, but we could not see anything that would suggest eye movement.

In addition, we also investigated the time-frequency profile of the filters— please see our answer to the next question (1a). Also our answers to comments (1b)-(1h) relate to the possible influence of eye-related activity.

1a. Saccade-related ERP profiles tend to have a positive peak near 0 ms followed by a negative peak around 20 ms. The attention-related ERPs using EEG, however, have key peaks at in the 100-200 ms range. Given this, the temporal profile of the filters may inform the arguments for and against eye movements contributing.

Thank you for the thoughtful suggestion. However, we kindly note that in this particular case the filters are not time-locked with the stimulus (we are not decoding a stimulus-following response as in traditional speech envelope reconstruction methods). That is, in our experiment, subjects continuously direct their attention to one ear, and for each x-second segment we determine the *direction* of attention, without relating/correlating the EEG to the stimulus waveform. We therefore don’t think a temporal profile as suggested would yield the desired result.

None the less, per the suggestion, we have calculated the frequency response of the filters in the convolutional layer. We did so in a grand-average fashion, similar to the topoplot that was added in the last revision (Figure 6 in the paper). That is, we first estimated the PSD of a single filter, averaged over all 64 channels, and subsequently averaged again over all five filters and over all runs and all window sizes. The result is a single, grand-average, magnitude response of the filters in the convolutional layer. The result is shown in Author response image 1. The relevant EEG-bands are shown on the figure, as well.

**Author response image 1. respfig1:** Grand-average temporal profile of the filters in the convolutional layer.

One can see that it is mostly the β band that is being targeted, which is also in correspondence with the results of the band-removal experiment that was also added in the previous revision (Figure 5 in the paper).

We feel that the temporal profile shown in Author response image 1 does not tell us anything new regarding the possibility that the model may in part be driven by eye movement, at least compared to what we already knew from the band-removal experiment. Even when relatively high frequency components are targeted, it does not automatically follow that these are saccades, or any other type of eye movement.

1b. Relatedly, if you found that the filters were tuned to γ band activity, this would suggest that small saccades are influencing performance. The fact that the network weights the β band as much as it does suggests that it may even like γ band more. On the other hand, if the filters are tuned to α or high δ, that would argue against saccades being the cause.

Please refer to our answer to (1a). In short, we do not feel that the fact that the filters are mainly tuned to the β band tells us much regarding the presence or non-presence of saccades.

1c. Your MWF algorithm should remove large gaze artifacts. However, even very small (but consistent) gaze changes could be responsible for some of the effects you see. You should also consider the literature on micro saccades and γ, and about whether small but consistent drifts of gaze during long trials contribute.

Thank you for the suggestion. We kindly note that a spatial filtering method such as MWF that attempts to remove large gaze artifacts will also remove smaller eye movements, as they originate from the same dipole as larger eye movements (the filter only uses spatial information).

1d. We are aware of your recent arXiv paper (Geirnaert et al) in which the CNN fails on another data set. Were subjects asked to fixate in that study, but not this? A better description of how subjects were instructed in the current study should be included, no matter what. Given the Geirnaert results, we think it is especially critical to figure out whether the results in the current paper really are attention effects in neural responses, rather than due to eye movement. It would be unfortunate to have to publish a correction if the results in the current study are attributed to attentional effects when they are actually due to gaze differences.

We agree that we could have been more clear regarding the instructions subjects received. We have added the following text to the “Experiment” section:

“The experiment was split into eight trials, each 6min long. In every trial, subjects were presented with two parts of two different stories. One part was presented in the left ear, while the other was presented in the right ear. Subjects were instructed to attend to one of the two via a monitor positioned in front of them. The symbol “*<*” was shown on the left side of the screen when subjects had to attend to the story in the left ear, and the symbol “*>*” was shown on the right side of the screen when subjects had to attend to the story in the right ear. They did not receive instructions on where to focus their gaze.”

The other dataset in Geirnaert et al. is the one published by Fuglsang et al. (2017). In this dataset subjects fixated on a crosshair. However, in pilot experiments with other datasets from our own lab we found that fixating on a point did not affect whether our DNN approach worked or not, so there must be another unknown difference between the Das et al., and Fuglsang et al. datasets.